# INTACT vs. FANS for Cell-Type-Specific Nuclei Sorting: A Comprehensive Qualitative and Quantitative Comparison

**DOI:** 10.3390/ijms22105335

**Published:** 2021-05-19

**Authors:** Monika Chanu Chongtham, Tamer Butto, Kanak Mungikar, Susanne Gerber, Jennifer Winter

**Affiliations:** 1Leibniz Institute of Resilience Research, Wallstr 7, 55122 Mainz, Germany; mchongth@uni-mainz.de; 2Institute of Human Genetics, University Medical Center of the Johannes Gutenberg University Mainz, Langenbeckstr. 1, 55131 Mainz, Germany; T.Butto@imb-mainz.de (T.B.); kamungik@uni-mainz.de (K.M.)

**Keywords:** FANS, INTACT, nuclei sorting, neuronal nuclei, ATAC-Seq, RNA-Seq

## Abstract

Increasing numbers of studies seek to characterize the different cellular sub-populations present in mammalian tissues. The techniques “Isolation of Nuclei Tagged in Specific Cell Types” (INTACT) or “Fluorescence-Activated Nuclei Sorting” (FANS) are frequently used for isolating nuclei of specific cellular subtypes. These nuclei are then used for molecular characterization of the cellular sub-populations. Despite the increasing popularity of both techniques, little is known about their isolation efficiency, advantages, and disadvantages or downstream molecular effects. In our study, we compared the physical and molecular attributes of sfGFP+ nuclei isolated by the two methods—INTACT and FANS—from the neocortices of Arc-CreERT2 × CAG-Sun1/sfGFP animals. We identified differences in efficiency of sfGFP+ nuclei isolation, nuclear size as well as transcriptional (RNA-seq) and chromatin accessibility (ATAC-seq) states. Therefore, our study presents a comprehensive comparison between the two widely used nuclei sorting techniques, identifying the advantages and disadvantages for both INTACT and FANS. Our conclusions are summarized in a table to guide researchers in selecting the most suitable methodology for their individual experimental design.

## 1. Introduction

The mammalian system consists of distinct and highly specific cell types that contribute to a given tissue’s function and, ultimately, promote the organism’s survival. Therefore, cell-specific studies have become highly relevant in uncovering such specific mechanisms. In recent years, many studies focused on developing different techniques to investigate the cells’ molecular properties, ranging from bulk to single-cell populations [1,2]. Fluorescence-activated cell sorting (FACS) has been widely used in a variety of molecular studies as a method to isolate populations using endogenous or external fluorescence tagging [3,4] since its introduction in the 1970s [5]. With the increasing interest in the genomic and epigenetic properties of specific cell types, the research focus has shifted to nuclei isolation techniques. Therefore, a variant of FACS was adapted, “fluorescence-activated nuclei sorting” (FANS), to isolate specific nuclei populations using nuclear fluorescence tags [6,7].

Although flow cytometry is a widely used technique to separate specific cells/nuclei populations, certain disadvantages have been associated with it. For instance, FACS/FANS requires high-cost equipment and specially trained people. These requirements can be an unaffordable obstacle, particularly for smaller labs. Additionally, it is challenging to sort morphologically complex cells such as cells from the central nervous system [8,9]. The high hydrodynamic stress forms one of the major challenges [10,11], as this could trigger cellular stress responses leading to a subsequent modification of different molecular profiles [8,12,13,14]. Apart from this, FACS/FANS also comes with a high time–cost factor to isolate rare cell populations at a multiscale level [1,9,15]. 

As an alternative, Deal and Henikoff [15] demonstrated a proof of principle for magnetic sorting of nuclei using genetically modified plants with the technique “isolation of nuclei tagged in specific cell types” (INTACT). Since then, this method has been adapted to model organisms such as *C. elegans, D. melanogaster* [16,17], and other mammalian systems [18,19]. Mo and colleagues introduced INTACT to sort activated neuronal (sfGFP+) populations in the mammalian brain [10,20]. Since then, the use of INTACT has been extended to distinct neuronal cells [18,19,21,22], cardiomyocytes [23,24,25], and hepatocytes [26,27]. INTACT and flow cytometry-based techniques from multiple literature data have already been reviewed in [1]. However, data of a systematic comparison, displaying the physical differences in the sorted nuclei, the overall efficiency of isolation, and subsequent molecular analyses between these two techniques are still lacking. Additionally, neither method has been tested or optimized for sorting small nuclei numbers, which are important for distinguishing between the numerous cellular subtypes’ expression profiles in the brain. For simplicity, FANS will be utilized as the default term to describe the work presented unless mentioned otherwise. 

In this study, we used a transgenic mouse model (Arc*^creERT2(TG/WT)^*.R26*^CAG-Sun1-sfGFP-Myc(M/WT or M/M)^*) to establish and compare the two techniques—INTACT and FANS—for the isolation of activated neuronal nuclei from small as well as bulk tissue samples. Using a modified protocol for nuclei isolation from small tissues [28], first, we examined the sorting processes’ speed and efficiency using different tissue types and sfGFP+ nuclei percentage. Then, we selected the neocortex (i.e., bulk tissue) to compare the physical (i.e., nuclei shape and intensity) and molecular (i.e., transcriptional profile and chromatin accessibility) aspects using low-input nuclei from the different sorting methods. We observed differences in the techniques in terms of both physical and molecular attributes. These observations are summarized in a simple informative overview to guide researchers towards a better experimental design for using cell-type-specific populations (to uncover unique cellular properties).

## 2. Results

### 2.1. Literature Review of INTACT and FANS/FACS Studies

To compare the methodologies of INTACT and FANS for cell-type-specific isolation, we first conducted a literature review to identify and summarize shared and distinct features of both techniques. We decided to limit the range of studies and focus on Mo et al.’s [10] citations, since we based our experimental comparisons on the same reporter line (R26*^CAG-Sun1-sfGFP-Myc^*). Additionally, this publication reports the first use of INTACT in the mammalian system, providing an essential landmark in the field. In the review, we included all studies that used INTACT/FANS (or both) regardless of the organismal system. We noticed that several studies used the term FACS while describing nuclei sorting and, therefore, we included them in the summary as FANS. The literature review comprised four stages including identification, screening, eligibility, and analysis (Figure 1A). We found that 31 publications used FANS, 18 used INTACT, and nine used both (Figure 1A). To track each technique’s dynamic use over time, we plotted the individual publications obtained from 2016 until 2020. Overall, we observed increased use of both methods each year (Appendix A). Of the studies under consideration, ~53% (31 out of 58) used FANS, ~31% used INTACT (18 out of 58), and ~16% used both (9 out of 58). 

Next, we looked at the subsequent molecular analyses used in conjugation with the investigated techniques. The four main fields of interest included RNA-seq, ATAC-seq, ChIP-seq, and DNA methylation analyses (Appendix A). Overall, we observed that FANS was applied more often than INTACT in combination with subsequent methodologies (Figure 1C). However, this was expected, as the use of flow-cytometry techniques precedes the application of techniques such as INTACT. Additionally, we observed an increase in the use of the molecular technique considered per year, most notably in RNA-seq and ATAC-seq, followed by a slight rise in ChIP-seq and DNA methylation analyses (Appendix A).

We further performed an analysis based on features such as the use of transgenics, tissue type, and organism. Due to the variability in data description, we opted to provide generalized information on the material used. For additional details of the specific tissues and quantities used, see Appendix A. In our results, we observed that the “brain” was the most frequently investigated tissue with 62% of all of the citations, followed by “plant tissue” (6%), “heart” (3.5%), and “liver” (2.4%) (Figure 1D, Appendix A). On the other hand, “*mus musculus*” was the most frequently used organism in the selected studies with 42% of the citations along with “*Homo sapiens*” (12%), “*Arabidopsis thaliana*” (5.8%), and “*Drosophila melanogaster*” (2.3%) (Figure 1D, Appendix A). The frequently used tissue and organisms are not surprising, as Mo et al. [10] investigated epigenetic signature changes occurring in mature neurons of transgenic mice. 

From our literature review, we observed an increased use of FANS and INTACT for cell-type-specific nuclei isolation. In terms of application and subsequent molecular analysis, FANS was used more frequently than INTACT. However, a comparative analysis of the two techniques concerning their suitability for downstream molecular analysis was lacking. Therefore, we decided to carry out such a comparative analysis starting with sorting efficiency using different brain sub-regions with varying amounts of nuclei and sfGFP+ percentages.

### 2.2. Assessing Differences in Speed and Sorting Efficiency between FANS and INTACT

To provide a comprehensive comparison between INTACT- and FANS-isolated nuclei, we used the tamoxifen (TAM)-dependent Arc*^creERT2 (TG/WT)^*.R26*^CAG-Sun1-sfGFP-Myc (M/WT or M/M)^* mouse line [10,29]. The mouse line expresses sfGFP on the nuclear membrane of specific stimulus-activated cells (expressing Arc) upon TAM injection. In our experiments, we used the “social interaction test” (SI, [30]) as the intended behavioral stimulus to obtain populations of sfGFP-positive (sfGFP+) nuclei, following TAM injection. Nuclei were isolated from the neocortices of 13–16 week old transgenic mice using the protocol from Chongtham et al. [28] (Figure 2A). Following isolation, nuclei were diluted with wash buffer and aliquoted into equal volumes to isolate the sfGFP+ nuclei using either INTACT or FANS (Figure 2B,C). Depending on the technique used, the sorted sfGFP+ nuclei were termed as INTACT-nuclei or FANS-nuclei.

It has been shown that INTACT is more efficient at processing larger amounts of samples compared to FANS because it requires less processing time [1]. Therefore, we first analyzed the speed of sfGFP+ nuclei isolation using INTACT or FANS under different conditions. We chose specific micro-dissected brain regions, which provided different nuclei yield and sfGFP+ nuclei percentages, in this experimental design (Table 1, Yield: sfGFP%; 73 × 10^4^/mL: 18.95; 60 × 10^4^/mL: 12.8; 140 × 10^4^/mL: 47.45, 136 × 10^4^/mL: 23, 200 × 10^4^/mL: 29.3 for nucleus accumbens, hypothalamus, pituitary, hippocampus, and neocortex, respectively). The regions were selected to represent diverse sample populations. 

The experiment was performed in a parallel run for all brain regions analyzed (technical duplicates). First, we noted the time required to obtain 5000 sfGFP+ nuclei from a single sample using FANS or INTACT. Next, the theoretical time needed to obtain large numbers (for example, 50k) was calculated using the unitary method. Here, we observed that a single sample’s processing speed was faster using FANS (Table 1, 1–3 min for FANS as compared to INTACT—35 min). However, when multiple samples were considered and the theoretical time was calculated, the FANS technique could be more time-consuming (100–200 min for FANS compared to INTACT—about 60 min).

Additionally, when using INTACT, despite the high overall purity of the separation, we observed differences in sorting efficiency, which was dependent on nuclei yield and sfGFP+ percentages. For example, for nucleus accumbens, where the nuclei yield was low, the yield of sfGFP+ nuclei was only 15%. In contrast, for pituitary with high nuclei yield and high sfGFP+ percentage, the extraction efficiency was approximately 40% (see Table 1, “yield”). However, when we applied the same protocol to the neocortex (high nuclei concentration and sfGFP+ percentage), the yield was approximately 86–90% (Table 1). The efficiency of nuclei extraction in FANS did not vary under these different conditions. 

Overall, our results indicate that the INTACT method might be preferable in cases of large sample numbers, where fast processing of nuclei is required. When rather accurate nuclei numbers are required for downstream processing, FANS may be better suited.

### 2.3. Quantification of Morphological Attributes: FANS- and INTACT-Nuclei in Comparison to INPUT-Nuclei

As in the previous section, 13–16 week old mice were subjected to an SI test 5 h after TAM injection to induce sfGFP expression on activated nuclei. Neocortices were dissected and nuclei were isolated. The experiments were repeated a total of three times on different experimental days using different biological replicates.

### 2.4. Structural and Optical Modifications Observed under Phase-Contrast Microscopy and Fluorescence Microscopy

Upon verifying the purity of the separated sfGFP+ populations (Appendix A), the nuclei morphology of FANS-nuclei and INTACT-nuclei were compared to unsorted nuclei (INPUT-nuclei) using phase-contrast microscopy and to sfGFP+ populations (INPUT-sfGFP nuclei) using fluorescence microscopy. For investigating structural modifications, we used phase-contrast microscopy to measure the “area” of FANS- and INTACT-nuclei in comparison to INPUT-nuclei (*n* = 100–200 nuclei; Figure 3A,B) from three different biological replicates. After sorting (<3 h), we detected a slight increase in the area and perimeter of FANS-nuclei compared to the INPUT-nuclei (*p* < 0.05, *p* < 0.0001), while INTACT-nuclei did not show a detectable change (Figure 3B, Appendix A).

Next, we quantified the “optical density” using the Trypan blue staining intensity, as the optical properties of a nucleus convey important information about the nucleus (including nuclear membrane integrity). Since the INTACT-nuclei beads affected the optical light transmission, we could only perform this analysis on FANS-nuclei compared to INPUT-nuclei. There was no significant difference between FANS-nuclei and INPUT-nuclei <3 h after sorting (*p* = 0.5397, n.s., Appendix A).

Using phase contrast microscopy, we could only compare sfGFP+ FANS- and INTACT-nuclei to a pool of sfGFP−and sfGFP+ INPUT-nuclei. This was suboptimal because most sfGFP+ nuclei from the neocortices are likely to be nuclei from excitatory neurons where Arc expression is abundant, whereas sfGFP−nuclei contain a mixture of all different cell types of the neocortex. Therefore, we used fluorescence microscopy for the next step to compare the sfGFP+ FANS- and INTACT-nuclei to sfGFP+ INPUT-nuclei. For these experiments, nuclei were embedded in the wells of a chambered coverglass (IBIDI) within 3 h or 4–6 h after sorting. Representative images of the INPUT-, FANS-, and INTACT-nuclei (<3 h) are shown in Figure 3C. Upon quantification, we observed a significant increase (*p* < 0.0001, *p* < 0.0001, *n* = 100–200 nuclei) in the size (area and perimeter) of sfGFP+ FANS-nuclei compared to that of sfGFP+ INPUT-nuclei (Figure 3D). The size of the INTACT-nuclei did not change significantly when compared to sfGFP+ INPUT-nuclei. The observations remain similar to that in phase-contrast microscopy. Thereafter, the increase in the size of FANS-nuclei was validated over a larger sample size using a macroscript [28] for faster processing (*p* < 0.0001, *n* = 300–400 nuclei, Appendix A, <3 h after sorting (A)). Already, 4–6 h after sorting the size of FANS-nuclei again decreased as shown in the Appendix A, *p* < 0.001. Interestingly, the increase in size (<3 h) was accompanied by a significant shift in the optical density of FANS-nuclei (only sfGFP+ populations considered) that showed a brighter pattern in the brightfield image (Appendix A, *p* < 0.0001 (NB: * these nuclei were without Trypan blue staining and, hence, exhibited the opposite pattern of that in Appendix A)). This finding suggests a possible change in FANS-nuclei’s osmotic pressure within a few hours of sorting, followed by subsequent shrinkage. 

Conclusively, the results indicate that FANS but not INTACT induces morphological changes in the sorted nuclei. The change in shape and optical density of FANS-nuclei, compared to INTACT-nuclei/INPUT-nuclei, could portray changes in chromatin dynamics and gene expression [31,32,33,34,35]. Therefore, we investigated the transcriptome and chromatin accessibility status in FANS- and INTACT-nuclei.

### 2.5. Transcriptional Differences of Low-Input RNA-Seq from FANS-Nuclei vs. INTACT-Nuclei

Following the qualitative assessment of INTACT and FANS on isolated nuclei, we focused on the sorted nuclei’s transcriptomic signatures. First, we examined two different RNA library preparation kits in order to select the most suitable one for nuclear RNA. The first was a polyA-based cDNA amplification kit (Clontech, Mountain View, CA, USA), while the second was a ribo depletion-based library preparation kit (NuGen Ovation Solo RNA-seq). Both kits dealt with low concentrations of RNA typically obtained from rare cell types or limited cell numbers. In this experimental design, RNA samples were isolated from 10,000 nuclei sorted with either INTACT or FANS using hippocampal sfGFP+ nuclei (RIN > 7) (Appendix A). Amplified cDNA libraries revealed similar cDNA distribution according to library preparation strategy (Appendix A). Following sequencing, we performed read alignment analysis to observe the distribution of mapped reads across the gene body. The read distribution was uniform across the gene body for Nugen compared to Clontech library preparation (Figure 4A). Additionally, 3’UTR distribution bias was observed in the Clontech library preparation as previously reported in poly(A) based strategies [3]. Using principal component analysis (PCA) [36], we observed that all samples clustered according to their isolation technique and library preparation strategy (Appendix A). To avoid 3’UTR distribution bias and since we were using low-input material isolated from the selected neuronal populations, we decided to use the Nugen Ovation Solo as our library preparation strategy for the following transcriptional analyses. 

Next, we examined the transcriptome differences between INTACT-and FANS-nuclei. For this comparison, we isolated RNA from neocortical sfGFP+ nuclei. We used the neocortex due to the ease of dissection and availability of higher number of isolated nuclei and sfGFP+ RNA quality was determined by the RNA integrity number (RIN). Isolation of RNA from low-input material results in variable RIN values that often do not reach the accepted RIN of ≥7 [3]. Therefore, we examined how RNA quality isolated from low-input nuclei influence the RNA-seq analysis of INTACT- and FANS-nuclei in order to identify potential transcriptional changes influenced by RIN values. Overall, we compared 14 replicates per technique deriving from three distinct biological sources (batch 1: *n* = 4 per technique; batch 2: *n* = 5–7 per technique; batch 3: *n* = 5–7 per technique) (Appendix A). Batch 1 contained samples with high RIN values ranging from 6.3–9.9, whereas batch 2 and 3 comprised samples with RIN values ranging from 4 to 7.5 (Appendix A). First, we evaluated alignment statistics to identify changes among the different batches (Appendix A). We observed no significant mapping variations between the samples with varying RIN values of both FANS and INTACT. Next, we compared the different batches and analyzed the samples according to the isolation technique. The PCA of all the samples discriminated the samples according to the batch of isolation (Figure 4B). When all batches were combined in PCA, only the high RIN batch (batch 1) separated the samples according to the isolation technique suggesting that the technique’s clustering became more evident with increasing RIN values (Figure 4B). Batch-specific PCA, revealed similar separation between INTACT and FANS samples within batch 1, whereas the discrimination within the lower RIN value batches (batches 2 and 3) was present but less distinct than in batch 1 (Appendix A).

To identify significant transcriptional differences between the techniques, we looked at differentially expressed genes (DEGs) between FANS and INTACT in each batch as well as among all the three batches. Within the high RIN value batch (batch 1), we identified 395 enriched and 422 depleted genes in INTACT compared to FANS, whereas 17 significantly enriched and nine depleted genes were identified in batch 2 and 3 enriched, and 66 depleted genes were identified in batch 3 (1.5 log2fold-change and *p*-value < 0.05; Appendix A). Using the obtained DEGs, we tested if the enriched and depleted genes were shared across the three distinct batches. In the INTACT-enriched genes, only 11 genes were shared across batches 1 and 2 (Figure 4C), whereas in the INTACT-depleted DEGs, we identified seven genes shared between batches 2 and 3, and six genes were shared between batches 1 and 3 (Figure 4D). The INTACT-depleted genes comprised various small nucleolar RNA (snoRNAs) (Figure 4D). Interestingly, Snora20 was the only significantly depleted gene in INTACT compared to FANS in all batches.

Finally, using batch 1 DEGs, we performed GO term enrichment analysis to identify potentially altered mechanisms between INTACT and FANS samples (Appendix A). Interestingly, the GO term for biological processes in INTACT-depleted genes were enriched for “regulation of transcription—DNA templated” (*p*-value = 5.25 × 10^−4^) and “mRNA processing” (*p*-value = 4.83 × 10^−5^) (Figure 4D), whereas INTACT-enriched genes were enriched for “transport” (*p*-value = 3.60 × 10^−7^) and “response to endoplasmic reticulum stress” (*p*-value = 5.59 × 10^−12^) (Figure 4E).

Overall, we did not observe major differences between INTACT and FANS when we compared the three distinct batches. However, batch-specific analysis revealed numerous DEGs between INTACT and FANS, especially for the high RIN value batch, which were associated with “transcriptional regulation” in FANS and “regulation of protein-related mechanisms” in INTACT. When comparing the distinct batches together, several Snor genes were depleted in INTACT compared to FANS, which reflects the possible changes in nuclear membrane integrity as observed in our qualitative assessment section. Following these observations, we next investigated whether nuclei sorting by INTACT or FANS leads to significant differences in the nuclei’s chromatin accessibility state.

### 2.6. ATAC-Seq Reveals Differences in Chromatin Accessibility State between FANS- and INTACT-Nuclei

Finally, to determine potentially significant differences in chromatin accessibility states between INTACT and FANS samples, we used an assay of transposable accessible chromatin coupled with high throughput sequencing (ATAC-seq) [37]. This technique allowed us to identify and compare the accessible chromatin region differences between these two techniques. We used three technical replicates from the same biological source to monitor the differences and similarities between the techniques. Following the initial analysis, we observed that all FANS and INTACT samples’ alignment rates were comparable. FANS samples had an average of 127,531 peaks (SD ± 3458), while INTACT samples had an average of 112,211 peaks (SD ± 12,222) (Figure 5A). Using a read distribution plot, we observed that the peak heights (i.e., reads assigned to particular peaks) at promoter/transcription start sites (TSS) were reduced in INTACT samples compared to FANS samples (Figure 5B). Similarly, peak distribution analysis showed a minor difference between FANS and INTACT, notably in peaks mapped to intergenic region and promoters of INTACT (33.76% and 21%, respectively) compared to FANS (30.17% and 22.23%, respectively) (Figure 5C,D). Additionally, we observed a large number of reads from INTACT samples mapped to the X-chromosome compared to FANS (Appendix A). Next, we performed differential peak analysis between FANS and INTACT samples. Here, we found 24,315 differential accessible regions (DARs) out of which 2578 DARs were at promoter/TSS sites of INTACT samples compared to FANS (Figure 5E and Appendix A). Gene browser tracks were used to illustrate the peak annotation and DARs between FANS and INTACT of selected gene promoter regions (Figure 5F). Further, selected promoter DAR were assigned to the nearest genes, and Gene Ontology (GO) analysis was performed to decipher the difference in biological processes between the DARs of the two techniques. INTACT gained-accessibility genes did not have significant enriched GO terms. On the contrary, INTACT loss-accessibility gene GO terms revealed multiple enriched GO-terms for biological processes, such as “transport” (*p*-value = 4.68 × 10^−5^) and “transcription—DNA templated” (*p*-value = 8.45 × 10^−7^), compared to FANS. Lastly, we performed motif enrichment analysis on the promoter regions of INTACT-loss and gained DARs (compared to FANS) in order to identify potential transcription factor motifs associated with technique-specific chromatin accessibility changes. For this analysis, we considered only promoter-associated motifs; however, the complete list of enriched motifs is presented in Appendix A. INTACT-gained accessibility revealed only the TATA-Box motif (*p*-value = 1 × 10^−2^) (Appendix A), whereas INTACT-reduced accessibility revealed six enriched motifs including: NRF (*p*-value = 1 × 10^−10^), ETS (*p*-value = 1 × 10^−9^), YY1 (*p*-value = 1 × 10^−8^), SP1 (*p*-value = 1 × 10^−7^), NFY (*p*-value = 1 × 10^−5^), and GFY-staf (*p*-value = 1 × 10^−2^) (Appendix A). These six motifs associated with INTACT-reduced accessibility (i.e., FANS-gained) have been previously labeled as “cardinal cis-regulatory motifs” suggested to be involved in regulation of chromatin structure by recruitment of secondary co-factors [38].

Overall, these results suggest that increased promoter accessibility of genes associated with transcriptional regulation is associated with FANS samples compared to INTACT. These observations agree with the transcriptional changes observed in the nucRNA-seq. It supports the hypothesis that changes in the physical properties of the sorted nuclei could lead to downstream molecular alterations.

## 3. Discussion

Increasing amounts of studies seek to characterize distinct cellular and rare sub-cellular populations. With the increasing interest in cell-type-specific epigenetic studies, novel techniques have evolved to isolate cell-type-specific populations with a shift of focus towards using nuclei [1]. Of these techniques, FANS has commonly been used to isolate cell-type-specific nuclei. Other methods, based on magnetic-based isolation principles, such as INTACT, have also been increasingly used. Although these two techniques have been interchangeably used, no study has performed an in-depth comparison of the two methods. This study provided a detailed qualitative and quantitative comparison between the two techniques, INTACT and FANS, using sfGFP+ nuclei. 

Our results from the literature review indicate a gradual increase in the use of both techniques within the past four years. The trend shows an increased interest in cell-type-specific studies (Figure 1). Additionally, we observed a common use of the organism “M*us musculus*” and “brain-related” tissue in the cited publications. These observations are not surprising, as [10] investigated epigenetic changes occurring in mature neurons using transgenic mice. We noticed several studies used the term FACS while describing nuclei sorting (i.e., FANS) [39,40,41,42,43]. It is reasonable that FACS will remain as general term for flow cytometry sorting; however, it is important to emphasize the material sorted and use appropriate terminology when required (e.g., FANS for nuclei sorting). Additionally, nine out of the 58 analyzed studies used both INTACT and FANS within the same study. Based on our results, we identified both physiological and molecular differences between the two techniques. Therefore, we recommend using only one method in the same study to avoid specific biases attributed by each technique. 

### 3.1. Physiological Differences between INTACT- and FANS-Nuclei

We observed that flow cytometry techniques (FACS/FANS) were the method of choice to isolate specific cells/nuclei. However, these techniques use high hydrodynamic pressure that could adversely affect the cells/nuclei [1,15,25]. Certain studies have reported adverse effects on cells’ viability and structure while using flow cytometry techniques [8,13]. Therefore, alternative methods of sorting, such as affinity purification methods, have been presented, such as magnetic bead sorting of astroglia/microglia for cell sorting [9,11]. INTACT has been introduced to separate cell-type-specific nuclei using magnetic beads. Another advantage of INTACT is the ease with which the sorted bead-bound nuclei can be concentrated into smaller buffer volumes compared to FANS [10]. Therefore, in our study, INTACT- and FANS-nuclei were compared in terms of nuclei structure/integrity and optical density. 

In our results, FANS-nuclei showed a significant increase in size compared to the INPUT nuclei, while INTACT-nuclei did not change significantly within 3 h of sorting. We hypothesize that the increase in FANS-nuclei size could be due to the high hydrodynamic pressure, shearing stress [8], or PBS-induced destabilization of membrane integrity during the sort [28,44]. Destabilized membrane integrity has been associated with flow cytometry as stated in previous studies [8,13]. We reason that the change in membrane integrity results in osmotic pressure alteration, which leads to an initial increase, followed by a size reduction after 4 h for FANS nuclei. Although Trypan staining did not reveal significant changes between Input-nuclei and FANS-nuclei within our recorded time, we speculate that differences might be observed if a longer incubation is performed. Accumulation of Trypan blue in nuclei has been associated with compromised membrane integrity [45,46,47], and the level of staining could indicate the extent of membrane permeability. 

Though Trypan blue staining showed no significant changes under phase-contrast microscopy, when FANS-nuclei were compared to the sfGFP+ nuclei population of the INPUT nuclei under AF7000 fluorescence microscope, a substantial change in optical density was observed in the brightfield. This could be due to the differences in nuclei properties of different cell types. The fluorescence microscopy data also revealed that reversal in the size of FANS-nuclei already occurs prominently after 4 h of the sort. All these structural and optical changes could portray possible molecular changes occurring in the FANS-nuclei compared to INTACT-nuclei.

### 3.2. Molecular Differences between FANS-and INTACT-Nuclei

We assessed the transcriptional and chromatin accessibility signatures following the microscopy observations, comparing FANS-to INTACT-nuclei. In order to get the best coverage of the nuclei contents, we examined different library preparation strategies for nuclear transcriptome. Previous studies compared distinct library preparation on whole-cell RNA [48,49] and nuclei [50] using polyA and rRNA depletion-based (Ribo-depletion) strategies. PolyA-based methods were reported to have better read coverage over exonic regions, and higher uniquely mapped reads than ribo-depletion-based techniques [51]. However, it relies mostly on “high”-quality intact RNA, as fragmentation can lead to 3’ coverage bias [52], which is also observed in our study. On the other hand, ribo-depletion based strategies have a higher percentage of mapped reads to intergenic and intronic regions compared to polyA-based strategies; hence, it requires a greater sequencing depth. Additionally, ribo-depletion based methods can be more useful for samples subjected to sequence fragmentation (lower RIN samples) [51]. We emphasize that even though RIN values provide valuable information regarding the RNA fragmentation state and concentration amounts, such values are unreliable considering low-input RNA and, particularly, nuclear RNA [3]. Coupling such low-RIN samples with ribo-depletion RNA-based or total-RNA library preparation strategies offers a suitable combination for nuclear RNA analysis.

To further explore the transcriptional differences between the two techniques, we performed transcriptional comparison for the FANS and INTACT RNA samples. Initial sequencing statistics revealed minor differences between INTACT and FANS. Using PCA clustering analysis, we observed that samples were clustered based on technique, with higher RIN samples leading to better clustering. We did not observe major changes in DEGs between INTACT and FANS when comparing all of the batches together (distinct biological sources). However, when comparing batch-specific DEGs, we identified numerous DEGs between FANS and INTACT, particularly for the batch with the high-RIN values. We reason that as the RIN value increases, the complexity of the RNA population increases leading to an elevated amount of DEGs as observed in our results. These observations agree with previous studies that have reported an association between degraded samples (low RIN) and reduced library complexity [53,54]. Additionally, we observed in our DEGs a large number of small RNA populations, particularly small nucleolar RNAs (snoRNAs). SnoRNAs were reported to be enriched in nuclei compared to the cytoplasm, which might explain the appearance of such RNA populations [55,56]. Further investigations are required to identify the association of such small RNA population with nucRNA-seq.

Additionally, we performed ATAC-seq in order to identify chromatin accessibility changes between the two techniques. Initial sequencing statistics revealed minor differences between INTACT and FANS. We observed reduced read mapping at promoter regions of INTACT samples compared to FANS, despite using the same starting material. This phenomenon was observed repeatedly in replicate ATAC-seq experiments (data not shown). Interestingly, we identified six motifs (NRF, ETS, YY1, SP1, NFY, and GFY-staf) associated with INTACT-reduced accessibility (i.e., FANS-gained) compared to the broad TATA-box motif observed in the INTACT-gained accessibility promoters. These six motifs have been previously labeled as “cardinal cis-regulatory motifs” which frequently co-occur and are suggested to be involved in regulation of chromatin structure by recruitment of secondary co-factors [38]. These results supplement our observations of depleted DEGs and reduced accessibility of GO term enrichment of INTACT compared to FANS that included “transcription, DNA-templated”, “RNA splicing”, and “mRNA processing”. These observations suggest that FANS samples contained an increase of transcriptional regulation associated genes compared to INTACT. We postulate that the physiological changes observed in FANS samples were sufficient to influence or reflect the respective molecular changes observed on transcriptome and chromatin accessibility states of the nuclei. It was previously reported that several forms of stress on the nuclear membrane, including mechanical force, could lead to chromatin alteration, which ultimately alters transcriptional activity [57,58,59]. Our results support these remarks, suggesting that nuclei sorting using the two different techniques could lead to different molecular signatures of the same biological materials.

Under our experimental setup, we observed differences between INTACT- and FANS-nuclei both on the physiological and the molecular levels. Both techniques have individual merits and demerits, as summarized in Table 2. We suggest that the choice of either INTACT or FANS for a particular study should be made based on the cell type investigated, starting tissue material, and subsequent molecular analyses. This is inherently supported by the molecular changes observed in our study. We strongly recommend selecting only one method to perform all the subsequent molecular analyses to prevent specific biases and inconsistencies between the different datasets. Overall, in terms of processing speed and costs for multiple samples, INTACT has an advantage over FANS. However, for INTACT protocol to be applied efficiently, the same amounts of starting material must be used. For smaller or higher amounts, the protocol has to be adjusted. Regardless of the starting concentration of nuclei, the FANS protocol need not be changed. In this aspect, FANS is advantageous over INTACT. The presented advantages and disadvantages of the compared techniques can help researchers to select the most suitable methodology for their individual experimental design. 

## 4. Materials and Methods

### 4.1. Literature Review of INTACT FANS/FACS Studies

The literature review was divided into four sections: identification, screening, eligibility, and analysis. Initially, in the identification stage, 335 citations of Mo et al.’s [10] were manually retrieved from the Google Scholar database. Following an initial assortment, 25 publications were removed due to the fact of repetition in the database, duplication in two different journals (peer-reviewed and non-peer-reviewed), or not eligible due to the language of publication (i.e., non-English). In the screening stage, we used an advanced search tool to find articles that included the terms “FANS”, “INTACT” or “FACS” on the remaining 310 articles. The articles were then classified further into “relevant” (79) (containing the searched terms) and “non-relevant” articles that consisted of “references” (156) and “reviews papers” (75) (Appendix A). The “relevant” publications used one/both of the techniques investigated, while “references” and “reviews” only cited Mo et al. [10] without using any of the techniques investigated in this study. The 79 “relevant” articles included 40 FACS, 12 FANS, 18 INTACT, and nine both (i.e., INTACT and flow-cytometry methods). We noticed that several studies used the term FACS while describing nuclei sorting. Therefore, we included them in the final summary as FANS (19/40 articles). To obtain a concise comparison of nuclei sorting techniques, we decided to exclude FACS-related articles (21/40 articles) that were sorting cells. Finally, 58 publications were considered for in-depth review and summary (Appendix A). The data were categorized into a table containing the following information: type of technique used, tissue, cell type, organism, sample size, subsequent analyses, and use of transgenic animals (Appendix A).

### 4.2. Animals and Ethics Statement

Thirteen to sixteen week old male and female Arc*^creERT2(TG/WT)^*.R26*^CAG-Sun1-sfGFP-Myc(M/WT or M/M)^* were used for this study [10,29]. Habituation in the experimental environment lasted for a week before performing tamoxifen (TAM) injections. All behavioral experiments were performed in accordance with the institutional animal welfare guidelines approved by the ethical committee of the state government of Rhineland-Palatinate, Germany, approved on 2 March 2017 (G-17-1-021).

### 4.3. Tamoxifen Injection and Behavioral Experiments

Tamoxifen (Sigma–Aldrich, St. Louis, MI, USA, T5648-1G) was dissolved in a corn oil:ethanol mixture (9:1) ratio at 42 °C with constant shaking at 500 rpm in light protected Eppendorf tubes (Amber). Animals were injected with tamoxifen at a dosage of 150 mg/kg 5 h before social interaction experiments to activate nuclei sfGFP expression under the control of the Arc promoter. Social interaction (SI) experiments were performed as described in [30]. Test mice were allowed to explore an open field box (40 cm wide × 40 cm depth × 40 cm length) that had an empty cylindrical cage at one side for 150 S in the habituation phase. Then, CD1 mouse was placed in the cylindrical cage and the test mouse was allowed to explore again in the test phase for another 150 S. Mice were sacrificed on the 3rd day following tamoxifen injections and SI to allow for optimal sfGFP expression on the nuclear membrane.

### 4.4. Nuclei Isolation from Different Brain Regions

Following behavioral experiments, the brain regions—neocortex, pre-frontal cortex, hippocampus, hypothalamus, pituitary, and nucleus accumbens—were dissected according to region specifications in the Allen Brain Atlas for mouse. Nuclei were isolated using the iodixanol-based gradient ultracentrifugation protocol for isolation of micro-dissected brain regions as described in Chongtham et al. [28]. The duration for high-speed ultracentrifugation (7820 rpm) for the neocortex was 12 min, while for the other brain regions, it was kept at 18 min. After the ultracentrifugation process, nuclei were collected from the 30–40% iodixanol layer interface.

### 4.5. sfGFP Positive Nuclei Separation

#### 4.5.1. INTACT

For sfGFP+ nuclei separation, Dynabeads (Life Technologies, Carlsbad, CA, USA, 10003D) were incubated with anti-GFP antibody (Life Technologies, Carlsbad, CA, USA, G10362) at 4 °C at a ratio of 50:20 (beads: anti-GFP) for downstream molecular biology experiments and at a ratio of 50:10 (beads: anti-GFP) for microscopy in order to decrease the number of beads bound to nuclei. The reduction of beads bound to a nucleus increases the visibility of bead-bound nuclei, which is important for microscopy analyses. The bead-antibody incubation was carried out in a cold room using an end-to-end rotator for 20 min to facilitate bead-antibody interaction in the solution. 

In the meantime, wash buffer (0.4% Igepal in homogenization buffer, [28]) was added to the nuclei solution, collected from the 30–40% iodixanol gradient layer (2:1). This was necessary to reduce the solution’s viscosity and remove the outer nuclear membrane before affinity purification (as mentioned in [60]) to facilitate anti-GFP binding to sun1sfGFP in the peri-nuclear membrane. The wash buffer-diluted nuclei solution was incubated with anti-GFP bead solution in the ratio of 40:1. Samples were then placed in an end-to end rotator in a cold room for 20 min. After the incubation period, samples were placed on a magnetic rack for 1 min, and the supernatant was removed, retaining the bead-bound sfGFP nuclei clusters. These clusters were washed ones with 500 µL wash buffer and twice with 250 µL wash buffer and subsequently collected in 100 µL wash buffer. For hemocytometer counting, proper dilutions were made so that not more than 200 bead-bound nuclei were present per large square. 

Following hemocytometer count, volumes that contained approximately 5000 nuclei for ATAC treatment and 10,000 nuclei for RNA extraction were calculated using a unitary method and collected accordingly.

#### 4.5.2. FANS

For FANS, the nuclei collected from the 30–40% iodixanol layer were diluted with wash buffer in the ratio 2:1 (similar to INTACT), as in Chongtham et al. [28] and Fernandez-Albert et al. [61], to reduce viscosity during sorting. Flow cytometry analysis and FANS were performed using a BD FACSAria III SORP (Becton Dickinson, Franklin Lakes, NJ, USA) equipped with four lasers (405 nm, 488 nm, 561 nm, and 640 nm) and a 70 µm nozzle. GFP expression was detected using the blue laser and a 530/30 BP filter, whereas DAPI was detected using the violet laser and a 450/50 BP filter. Prior sort, 10,000 total events were recorded, and a gating strategy was applied: first, nuclei were gated according to their forward-and side-scatter properties (FSC-A/SSC-A), followed by doublet exclusion using SSC-A and SSC-W. Nuclei were then gated according to their DAPI expression. GFP expression was used as a sorting gate. Sorted nuclei were collected in 1.5 mL Eppendorf tubes, containing the optimal buffers for downstream analyses. The flow cytometry analyses were performed using the BD FACSDiva 8.0.2 Software or FlowJo (v10.6 or higher). 

#### 4.5.3. Purity Analysis of INTACT- and FANS-Nuclei

For purity analysis of INTACT-nuclei, the solution was diluted with wash buffer in a ratio of 4:1 before flow cytometry to reduce the density of bead-bound nuclei. For FANS-nuclei, the sorted sfGFP+ nuclei were collected in wash buffer to keep the experimental conditions similar to that of INTACT. The nuclei purity analysis was performed using BD FACS Aria III (Becton Dickinson, Franklin Lakes, NJ, USA). Data were analyzed with FlowJo (v.10.6 or higher, Tree Star, Ashland, OR, USA) from at least 50 single DAPI-positive nuclei. 

### 4.6. Parallel Processing of INTACT- and FANS-Nuclei

For comparing the processing speed, different brain regions (nucleus accumbens, hypothalamus, pituitary, prefrontal cortex, neocortex; see Table 1) with varied combinations of different sfGFP+ percentages sfGFP+ nuclei yield per volume were used. For FANS, duration of sorting to obtain 5k sfGFP+ nuclei were determined for each sample type. INTACT was performed as usual for all sample types. Bead-bound nuclei were then counted using a hemocytometer and the volume that contained 5k bead-bound nuclei was calculated. The duration for sorting 50 k nuclei using both techniques was then calculated using unitary method. For INTACT, the efficiency of sorting (yield) was calculated indirectly by using the percentage of sfGFP+ nuclei that remained in the INTACT supernatant compared to the percentage of sfGFP+ nuclei in the original solution.

### 4.7. Microscopy

For phase-contrast microscopy, 10 µL of the nuclei:Trypan mixed in a 1:1 ratio was loaded into the hemocytometer. Light intensity was kept the same for all groups. Images were obtained using Leica DM-IL inverted microscope with 40×/0.5, air objective. For fluorescence microscopy, images were obtained using a Leica AF7000 widefield microscope equipped with a Hamamatsu-Flash4-USB3-101292 camera, an LED lamp (Sola Light Engine, SE 5-LCR-VB, Lumencor, Beaverton, OR, USA) and LAS X software (Institute for Molecular Biology, Mainz, Germany). Images were acquired with a HC PL FLUOTAR L 20/0.40 or a 40/1.1 objective lens using the same settings for each image. Images visualizing GFP fluorescence were acquired using an L5 filter (BP 480/40, FT505, BP 527/30), and DAPI was imaged using an A4 filter (BP 360/40, FT400, BP 470/40). Images were analyzed using FIJI (v 1.51 h) as in Chongtham et al. [28]. Manual analysis was performed by drawing an ellipse in the area of the image encompassing the nucleus under observation while automatic analysis was performed using a macroscript [28]. Statistical analyses were performed using Student’s unpaired *t*-test (assumption: equal SD between populations) in GraphPad Prism (8.4.2).

### 4.8. RNA Isolation and nucRNA-Seq Library Preparation

Ten thousand INTACT- or FANS-nuclei were collected in 100 μL of RLT buffer followed by flash freezing. RNA was purified using the RNeasy Micro kit (Qiagen 74004, Hilden, Germany) following the manufacturer’s instructions. For the RNA-seq library comparison, we used two different protocols: a ribo-depletion based method, using the Ovation^®^ SoLo RNA-Seq System (NuGEN M01406v2, Redwood City, CA, USA) and a polyA enrichment-based protocol, using SMART-Seq^®^ v4 Ultra^®^ Low Input RNA Kit for Sequencing (Clontech 634894, Mountain View, CA, USA). 

NuGEN’s Ovation SoLo protocol: NGS library preparation was performed following NuGEN’s standard protocol (M01406v2). Libraries were prepared with a starting amount of 1.5 ng and amplified in 14 PCR cycles. *Clontech’s SmartSeq protocol*: Clontech’s SMART-Seq v4 Ultra Low Input RNA Kit (112219) was used for cDNA generation from 1.5 ng of total RNA, following the manufacturer’s recommendations. cDNA was amplified in 11 cycles of LD-PCR. The resulting cDNA were sheared using an S2 focused ultrasonicator (Covaris, Woburn, MA, USA) with the following parameters: 20% duty cycle; 0.5 intensity; 50 cycles/burst; 20 °C; 60 s. The NGS library preparation was performed with 3.16 ng of sheared cDNA with NuGEN’s Ovation Ultralow System V2 M01379 v5. Libraries were amplified in 11 PCR cycles. 

Both NuGEN and Clontech libraries were profiled in a High Sensitivity DNA Chip on a 2100 Bioanalyzer (Agilent technologies, Santa Clara, CA, USA) and quantified using the Qubit dsDNA HS Assay Kit, in a Qubit 2.0 Fluorometer (Life technologies, Carlsbad, CA, USA). All eight samples (4 NuGEN and 4 Clontech) were pooled together in equimolar ratio and sequenced on 1 NextSeq 500 Midoutput Flowcell, PE for 2 × 72 cycles plus 16 cycles for the index read and five dark cycles upfront. These dark cycles were included to avoid having unbalanced first bases of Read 1, introduced in the reverse transcription step of the NuGEN’s Ovation SoLo protocol. Consequently, there was no need to trim the reads for data analysis. For INTACT and FANS nuclear RNA-seq comparison, libraries were prepared using NuGen Ovation SoLo RNA-Seq System following NuGen’s standard protocol. Libraries were prepared with a starting amount of 1.23 ng and amplified in 13 PCR cycles. All samples (INTACT and FANS) were pooled in equimolar ratio and sequenced on 1 NextSeq 500 Highoutput Flowcell, SR for 1 × 70 cycles plus 16 cycles for the index read and five dark cycles upfront. All RNA-seq library preparations and sequencing were performed by the Genomic Core Facility from the Institute of Molecular Biology (IMB, Mainz, Germany).

### 4.9. RNA-Seq Data Analysis

FANS and INTACT Nugen Ovation RNA Solo library single end sequenced data quality assessment was performed using FASTQC (v. 0.11.8). Read alignment was done using STAR aligner (v2.7.1a) [62] to the *Mus musculus* genome (mm10) UCSC annotations with default parameter. Further, duplicates were removed using UMI (Unique Molecular Identifier) introduced by Nugen Ovation RNA Solo library. Uniquely mapped reads were retained in the output BAM file. Samtools(v1.7) [63] was used to sort and index mapped files. Read count per gene was calculated using HTSeq (v0.11.1) [64]. Normalization and differential expression analysis were conducted using the DESeq [63] Bioconductor package with an FDR rate of 0.05. The DAVID database was used for gene ontology analysis. 

### 4.10. ATAC-Seq Library Preparation

The ATAC-seq was performed in three technical replicates per technique as previously described [37]. Briefly, 50,000 INTACT or FANS nuclei were collected in 50 μL cold lysis buffer (10 mM Tris-HCl, pH 7.4, 10 mM NaCl, 3 mM MgCl_2_ and 0.1% IGEPAL CA-630) and centrifuged at 750× *g* for 15 min using a cold centrifuge (4 °C). Immediately after centrifugation, the pellet was resuspended in the transposase reaction mix (25 μL 2× TD buffer, 2.5 μL transposase (Illumina FC-121–1030, San Diego, CA, USA) and 22.5 μL nuclease-free water). The transposition reaction was carried out for 30 min at 37 °C. Following transposition, the sample was purified using a Qiagen MinElute kit (Qiagen, 28006, Hilden, Germany) following the manufacturer’s instructions. After purification, the library was amplified using 1× NEBnext PCR master mix (NEB M0541, Ipswich, MA, USA) and 1.25 μM of custom Nextera PCR primers 1 and 2 (See [37]). An optimization qPCR quantification was performed as described in Buenrostro et al. [37]. The following PCR conditions were used: 72 °C for 5 min; 98 °C for 30 s; thermocycling at 98 °C for 10 s, 63 °C for 30 s, and 72 °C for 1 min. After 11–12 cycles of PCR amplification, samples were further purified using Qiagen MinElute kit. To remove primer dimers, samples were further purified using AMPure beads XP (Beckman Coulter, Brea, CA, USA) with a ratio of ×0.9 of beads to samples. The purified samples were then analyzed in bioanalyzer (Agilent) and sequenced on NextSeq 500 Highoutput Flowcell (150-cycles PE for 2 × 75bp). Sequencing was performed by the Genomic Core Facility from the Institute of Molecular Biology (IMB, Mainz, Germany).

### 4.11. ATAC-Seq Data Analysis

The ATAC-seq data quality check was performed using FASTQC (v. 0.11.8). Further, adaptors were removed using Trimmomatic (v0.39) [65]. Paired-end ATAC-seq reads were mapped to *Mus musculus* genome (mm10) UCSC annotations using Bowtie2 (v2.3.5.1) with default parameters. Properly paired end reads with high mapping quality (MAPQ ≥ 10) were retained in analysis with the help of Samtools (v1.7). Next, using Picard tools MarkDuplicates utility, duplicates were removed. ATAC-seq peaks were called using MACS2 (v2.1.1.20160309) [66] and visualized with UCSC genome browser. Peaks that were reproducible in all samples were considered for differential accessible region analysis. Differential accessible region analysis was performed using DESeq (*p*-value cutoff of 0.05). Further, peaks were annotated using the annotatePeaks.pl utility of HOMER. For motif enrichment analysis findMotifGenome.pl utility (HOMER) was used. 

## Figures and Tables

**Figure 1 ijms-22-05335-f001:**
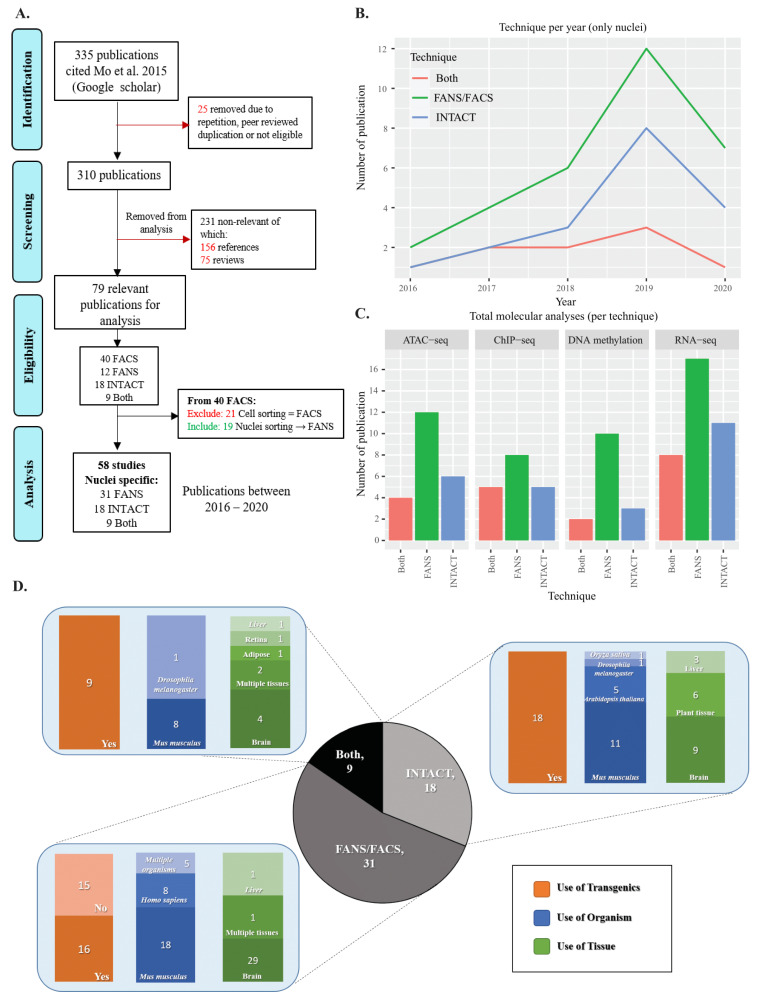
Literature review of Mo et al.’s (2015) citations: (**A**) literature review workflow for assessing the usage of INTACT and FANS; (**B**) publication trend of individual techniques per year; (**C**) use of subsequent sequencing techniques per year of publications for INTACT, FANS or both; (**D**) overview of the use of transgenics, organism, and tissue per technique (see Appendix A for additional information).

**Figure 2 ijms-22-05335-f002:**
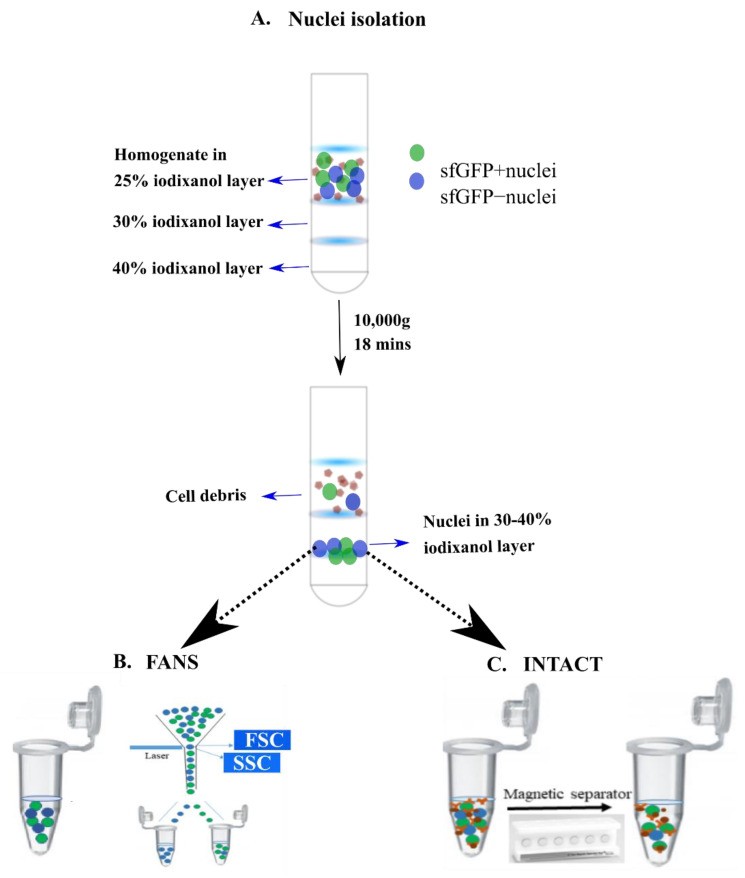
FANS vs. INTACT: Procedural differences and purity estimation: (**A**) nuclei extraction via ultracentrifugation (adapted from Chongtham et al. (2020)); (**B**) nuclei sorting using FANS; (**C**) magnetic separation of bead-bound nuclei using INTACT.

**Figure 3 ijms-22-05335-f003:**
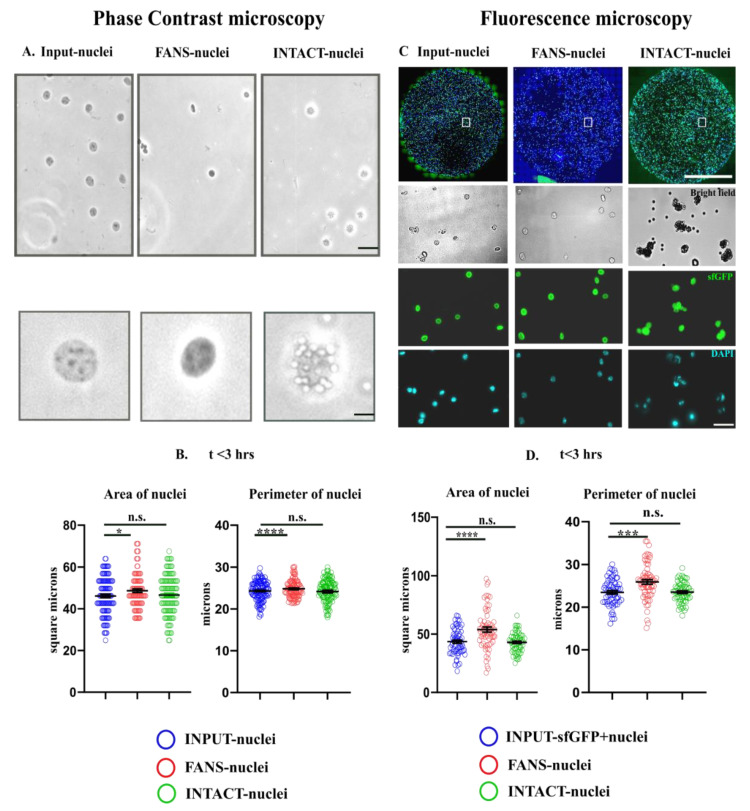
Morphological comparisons between FANS- and INTACT-sorted nuclei. Nuclei were isolated from the neocortices of 13–16 week old mice and sfGFP+ nuclei were sorted using FANS or INTACT. (**A**) For phase contrast microscopy, sorted nuclei were observed within 2–3 h of the sort. Representative images of INPUT-, FANS- and INTACT-nuclei. Scalebar: 25 µm. (**B**) Nuclei area and perimeter after sorting with FANS or INTACT. Both parameters in FANS- but not INTACT nuclei (n.s. (*p* > 0.05) increased significantly compared to the INPUT-nuclei (*p* < 0.05 (*), *n* = 100–200 nuclei). Scalebar: 5 µm, Air Objective, 40×. (**C**) For fluorescence microscopy, sorted nuclei were embedded in IBIDI chambers within 3 h or within 4–6 h of the sort. Representative images of INPUT-, FANS- and INTACT-nuclei (within 3 h of the sort) in the wells of IBIDI chambers and the representative enlarged images showing brightfield, GFP, and DAPI fluorescence along with the magnetic beads (2.8 µm diameter) showing a green autofluorescence (for INTACT) are shown. Scale bar: 2 mm (inset), scale bar—25 µm, water objective, 20× and 40×. (**D**) Analysis of nuclei area embedded in the IBIDI chambers within 3 h after sorting manually. Significant increase in the size of FANS-nuclei <3 h of sorting (nuclei area and perimeter) compared to the sfGFP+ INPUT-nuclei (*p* < 0.001 (***), *p* < 0.0001 (****), *n* = 100–200 nuclei) while no significant changes in INTACT-nuclei.

**Figure 4 ijms-22-05335-f004:**
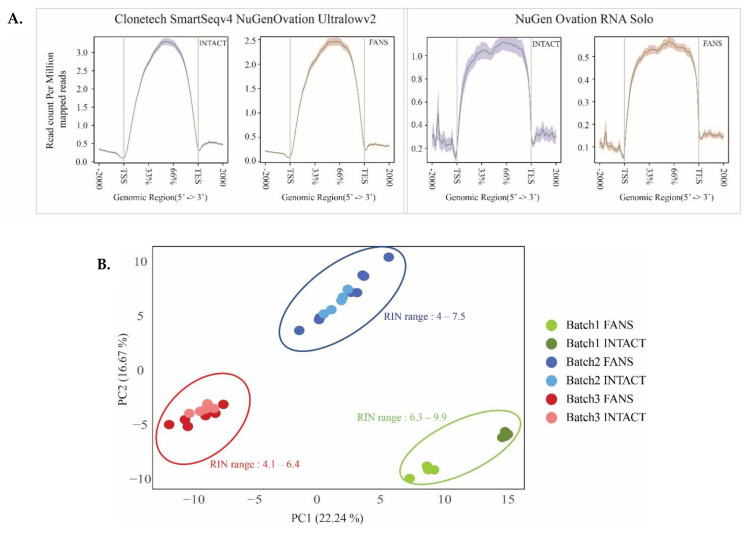
Transcriptional comparison between FANS- and INTACT-nuclei with distinct RIN values: (**A**) FANS/INTACT-mapped reads across the gene body based on library preparation strategy; (**B**) combined PCA from the three batches with distinct RIN values; (**C**) Venn diagrams of shared upregulated genes in INTACT compared to FANS among the 3 batches (1.5-fold change, *p* < 0.05); (**D**) Venn diagrams of shared downregulated genes in INTACT compared to FANS among the 3 batches (1.5-fold change, *p* < 0.05); (**E**,**F**) GO term enrichment analysis of INTACT-depleted (**E**) genes and enriched (**F**) compared to FANS. (Data represent depleted and enriched in INTACT compared to FANS).

**Figure 5 ijms-22-05335-f005:**
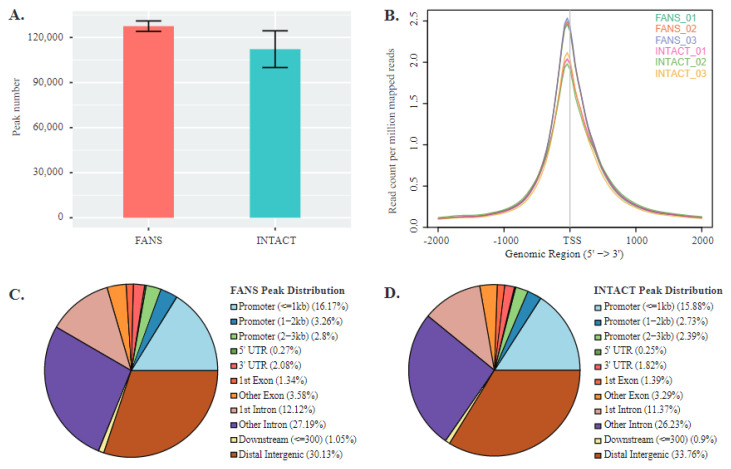
Chromatin accessibility comparison between FANS- and INTACT-nuclei: (**A**) average peak number in FANS and INTACT; (**B**) mapped reads coverage at transcriptional start site (TSS) in FANS and INTACT; (**C**,**D**) peak distribution across distinct genomic regions of FANS (**C**) and INTACT (**D**); (**E**) volcano-plot of differentially accessible regions (DARs) between INTACT and FANS; (**F**) GO term enrichment analysis for INTACT loss-accessibility genes compared to FANS; (**G**) genome browser tracks of selected genes Arc (IEGs), Gapdh (positive control), Sucnr1 (Gained accessibility in INTACT), and Prss34 (Gained accessibility in FANS).

**Table 1 ijms-22-05335-t001:** Processing speed of sfGFP+ nuclei separation using FANS and INTACT (mean of biological replicates).

Brain Region\Property Tested	Nuclei Yield (×104/mL)	Total Volume (μL)	Input Nuclei sfGFP+%	Actual 5k Nuclei/Sample (mins)		Theoretical 50k Nuclei/10 Samples (mins)		sfGFP+ Nuclei Yield from INTACT		
				FANS	INTACT	FANS	INTACT	GFP% Supernatant	Yield %	Purity
Small–medium brain regions										
Nucleus accumbens	73	300	18.95	2	35	200	60	16.5	15	98.5
Hypothalamus	60	400	12.8	3	35	300	60	7.6	44	98.2
Pituitary	140	400	47.45	1	35	100	60	28.9	40	99.1
Hippocampus	136	400	23	1.5	35	150	60	12.3	50	99.8
Large brain regions										
Neocortex 1	>200	900	29.3	-	-	-	-	3.6	87	98.5
Neocortex 2	>200	900	26.7	-	-	-	-	3.7	86	93.6

**Table 2 ijms-22-05335-t002:** Qualitative and quantitative differences between INTACT and FANS.

Parameter Compared	INTACT	FANS
**Experimental approach**	➢ Affinity purification-based approach of nuclei isolation	➢ Flow cytometry-based approach of nuclei isolation
**Processing speed**	➢ Relatively high processing speed. ➢ Processing of multiple samples ➢ Reduced experimental waiting time for multiple samples	➢ Relatively low processing speed. ➢ One sample collection at a time ➢ Increased experimental waiting time (if many samples considered)
**Quantification accuracy**	➢ Decreased accuracy compared to FANS ➢ Quantification based on manual hemocytometer counting	➢ Higher accuracy compared to INTACT
**Tissue/cellular amounts requirement**	➢ Protocol needs to be modified for different tissue amounts	➢ Protocol remains the same for any type of tissue amount (from low to high)
**Effect on nuclear structure**	➢ Physiological changes of isolated nuclei undetectable under the experimental setup	➢ Physiological changes detectable under the experimental set-up
**Transcriptional alterations**	➢ Minor inter-variability between samples ➢ Enriched genes associated with “transport” and “response to ER stress”	➢ Considerable inter-variability between samples ➢ Enriched genes associated with “regulation of transcription and “mRNA processing”
**Chromatin accessibility alteration**	➢ Accessibility peaks associated with intergenic compared to FANS ➢ Reduced accessibility in promoters of gene associated with “transcription- DNA template”, “mRNA-processing” and cellular response to DNA damage stimulus”	➢ Accessibility peaks associated with promoters compared to INTACT ➢ No significant reduced accessibility GO terms compared to INTACT
**Cost**	➢ Cost-effective	➢ Expensive equipment required

## Data Availability

The accession numbers reported in this paper are RNA-seq and ATAC-seq (GSE165644). The UCSC browser session link of ATAC data is http://genome.ucsc.edu/s/kanak/INTACTvsFANS_ATACseq. Flow cytometry data were deposited in the FlowRepository (Repository ID: FR-FCM-Z3FX; URL: http://flowrepository.org/id/RvFr7OlCuBDNO3jJvcCSmzBtpm8kB4awZ171aa6EYpEWhRexLS53Huns0tPx3RDR.

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
