# Peer review of "INTACT vs. FANS for Cell-Type-Specific Nuclei Sorting: A Comprehensive Qualitative and Quantitative Comparison"

_ijms, 2021, doi:10.3390/ijms22105335_

Round 1

Reviewer 1 Report

In “INTACT vs. FANS for cell-type-specific nuclei isolation: A comprehensive qualitative and quantitative comparison” the authors want to explore the differences between two methods using the same biological sample.

This is useful and worth pursing, however, the authors have overlooked a major issue by only focusing on the differences: all of the samples from both methods should be more similar than different. In fact, I am sure that if the authors were working on a different paper, the small differences would have been irrelevant, and only the similarities would have been focused on – each method removing experimental bias from the other.

By doing so, the authors claim to have found measureable, reproducible biological effects on FANS sorted nuclei. This is a big prediction, which was left untested.

In order to remove this bias, the authors need to address the similarities just as much so, as the differences. You will always find differences - if you focus on them.

To fix these issues I request:

  • The authors should look at and report the similarities between the two methods. This should compose Figure 4 and encapsulate, if still justified, the main argument. Venn diagrams, statistical validation, etc. The authors should come up with what is important. After doing that, they should move their PCA in the supplement to the main figure, as that is their evidence that the FANS and INTACT data can be split, since PCA is an analysis of variance. I am not sure if there is a needed statistical value for this – but a RMSD of the spread would likely better than just “the data are more variable”.
    • Actually, the Supp5A shows me that the two methods, for all classes are very similar. Only group I, and almost group III, separate cleanly by method and is the major variance carrier (PC1). Group II shows that they cluster, but do not support the major variance hypothesis by method.
    • Figure 4 is bleeding over the page, it is too big. Panel A shows no differences by nuclear collection method, but by RNA extraction etc. This is brilliant, as we see that post-processing methods also will have experimental error associated with them. Maybe this can used by the authors – but this is only valid if the noise is a consistent one.
  • Maybe Figure 5 should come before Figure 4? I see more similarities except for the volcano plot, but this is only valid if those samples should be pooled. If there are only random differences, then a randomization experiment should have been conducted, although here I am inclined to trust the PCA.
  • Figure 3 should only include comparative data. C, F, G, and E (right panel) do not include INTACT data. As most of the samples were finished ≤ 3 hrs, it makes little sense to show 24 hrs. I did not even get if the authors would/did take samples for sequencing after 24 hrs? This worried me as it feels like the authors are looking for differences. Again, the methods should be relatively equivalent with the same material. Move all that data to the supplement, and move the comparative data that is in the supplement (SuppF2) to the main figure.
  • Personally I am not a fan of GO categories. But, that aside, the authors main conclusion is that their methods reproduce particular biological biases in the analysis: transport, transcription, phenotypical changes. I would be more inclined to believe the authors if they were able to reproduce this effect in other datasets. Of the 9 studies that used both methods, was there at least one that was done under similar conditions with identical material? The authors could look at those studies to see if they can reproduce their findings. Otherwise, it could be that what they have observed is only valid for their particular material under their particular experimental conditions.

Figures

I hope all of the figures can be or are prepared in a higher resolution. The quality is quite poor, for all of them. Is that just the compressed version for review?

Table 1 - is confusing to read. Can you add other colors/grey scales to add column grouping and delimitation?

Figure 2 - FACS /beads - is misleading. Where are the non-positive nuclei? Where is the gating strategy?

Figure 2 is misleading and I hope this is not what the authors did; the cytometry documentation is incomplete and not present in Figure 2 or in the supplementary material. I don’t expect all of the cells are expressing GFP, therefore, the proper plot is one of DAPI vs GFP and before and after sorting, clearly showing how they identified the GFP+ nuclei.  Same with the beads, b.t.w. which I think is a great idea – but also – it needs to be done for both +/- GFP populations…a before and after. Plus, a supplemental figure of the entire gating regions for FACS and analysis should be in the supplementary information. Examine these guidelines: https://flowrepository.org/faq?topic=e3. For the beads we also need beads alone and beads with DAPI.

Again: before and after – not just re-analysis. Plus +/- GFP populations should be shown as well using DAPI. Not just scatter – which is okay – but not critically informative as the initial sample is likely not so clean.

  1. the authors did realize they can count the nuclei clusters with flow cytometry instead of using the haemocytometer, right?

Figure 3 – data do not support the major conclusions

Anything above 3 hours is only for FANS and not INTACT – therefore, the authors may not discuss this anywhere as a comparative result. Therefore the conclusion on line 219 is wrong. I did not understand would anyone want to wait for 24 hrs before processing nuclei, especially if they are not fixed. And this is more confusing as from Table 1 the longest sort was only 5 hours (300 mins).

Section 2.4 – FANS and INTACT are not treatments in the classical sense

up-regulated? down-regulated?  -> better is enriched/depleted or more/less present compared to reference. Aren’t all of the samples from the same biological source? There is no up or down really so the expression should be changed and please make sure the reference sample is clearly indicated. I think sometimes it was, sometimes it wasn’t. If it will be consistent – make a statement first which the reference will be, and then continue.

Author Response

Reviewer 1)

In “INTACT vs. FANS for cell-type-specific nuclei isolation: A comprehensive qualitative and quantitative comparison” the authors want to explore the differences between two methods using the same biological sample.

This is useful and worth pursing, however, the authors have overlooked a major issue by only focusing on the differences: all of the samples from both methods should be more similar than different. In fact, I am sure that if the authors were working on a different paper, the small differences would have been irrelevant, and only the similarities would have been focused on – each method removing experimental bias from the other.

By doing so, the authors claim to have found measureable, reproducible biological effects on FANS sorted nuclei. This is a big prediction, which was left untested.

In order to remove this bias, the authors need to address the similarities just as much so, as the differences. You will always find differences - if you focus on them.

To fix these issues I request:

  • The authors should look at and report the similarities between the two methods. This should compose Figure 4 and encapsulate, if still justified, the main argument. Venn diagrams, statistical validation, etc. The authors should come up with what is important. After doing that, they should move their PCA in the supplement to the main figure, as that is their evidence that the FANS and INTACT data can be split, since PCA is an analysis of variance. I am not sure if there is a needed statistical value for this – but a RMSD of the spread would likely better than just “the data are more variable”.

We agree with the reviewer. This is indeed an important point. We have now modified the section 2.4 (including the figure) to display both, similarities and differences between INTACT and FANS. To show the similarities between all samples of the three batches we have now included the PCA plot and Venn diagrams in Figure 4B-D.

  • Actually, the Supp5A shows me that the two methods, for all classes are very similar. Only group I, and almost group III, separate cleanly by method and is the major variance carrier (PC1). Group II shows that they cluster, but do not support the major variance hypothesis by method.

The PCA plot is now displayed as Figure 4B in the manuscript.

  • Figure 4 is bleeding over the page, it is too big. Panel A shows no differences by nuclear collection method, but by RNA extraction etc. This is brilliant, as we see that post-processing methods also will have experimental error associated with them. Maybe this can used by the authors – but this is only valid if the noise is a consistent one.

We have now re-arranged this figure.

  • Maybe Figure 5 should come before Figure 4? I see more similarities except for the volcano plot, but this is only valid if those samples should be pooled. If there are only random differences, then a randomization experiment should have been conducted, although here I am inclined to trust the PCA.

As stated above we have modified section 2.4 to show not only the differences but also the similarities between INTACT and FANS in the nucRNA-seq data. By this, we would like to maintain the order and first report similarities/differences in gene expression and then in chromatin accessibility.

  • Figure 3 should only include comparative data. C, F, G, and E (right panel) do not include INTACT data. As most of the samples were finished ≤ 3 hrs, it makes little sense to show 24 hrs. I did not even get if the authors would/did take samples for sequencing after 24 hrs? This worried me as it feels like the authors are looking for differences. Again, the methods should be relatively equivalent with the same material. Move all that data to the supplement, and move the comparative data that is in the supplement (SuppF2) to the main figure.

We agree that the 24 hrs is confusing. After 24 hrs samples would definitely not been taken for sequencing, and we therefore decided to remove the 24 hrs data from the manuscript.

In addition, we have moved the comparative data to the main figure.

  • Personally I am not a fan of GO categories. But, that aside, the authors main conclusion is that their methods reproduce particular biological biases in the analysis: transport, transcription, phenotypical changes. I would be more inclined to believe the authors if they were able to reproduce this effect in other datasets. Of the 9 studies that used both methods, was there at least one that was done under similar conditions with identical material? The authors could look at those studies to see if they can reproduce their findings. Otherwise, it could be that what they have observed is only valid for their particular material under their particular experimental conditions.

Unfortunately, the conditions (e.g. tissue type, combination of sorting technique with sequencing approach, model organism) in these studies differed so much from our approach that we cannot use these studies for such an analysis. 

Figures

  • I hope all of the figures can be or are prepared in a higher resolution. The quality is quite poor, for all of them. Is that just the compressed version for review?

We have now adjusted the figures’ quality.

  • Table 1 - is confusing to read. Can you add other colors/grey scales to add column grouping and delimitation?

We have now modified the colors and linings as suggested.

  • Figure 2 - FACS /beads - is misleading. Where are the non-positive nuclei? Where is the gating strategy?

We have now included all the plots in Supplementary Figure 2.

  • Figure 2 is misleading and I hope this is not what the authors did; the cytometry documentation is incomplete and not present in Figure 2 or in the supplementary material. I don’t expect all of the cells are expressing GFP, therefore, the proper plot is one of DAPI vs GFP and before and after sorting, clearly showing how they identified the GFP+ nuclei.  Same with the beads, b.t.w. which I think is a great idea – but also – it needs to be done for both +/- GFP populations…a before and after. Plus, a supplemental figure of the entire gating regions for FACS and analysis should be in the supplementary information. Examine these guidelines: https://flowrepository.org/faq?topic=e3. For the beads we also need beads alone and beads with DAPI.

Again: before and after – not just re-analysis. Plus +/- GFP populations should be shown as well using DAPI. Not just scatter – which is okay – but not critically informative as the initial sample is likely not so clean.

As stated above we have now included the plots as Supplementary Figure 2.

  • the authors did realize they can count the nuclei clusters with flow cytometry instead of using the haemocytometer, right?

We were aware of this possibility. Under our experimental conditions (e.g. for observing bead bound nuclei in INTACT experiments) we decided to use the haemocytometer.

  • Figure 3 – data do not support the major conclusions

Anything above 3 hours is only for FANS and not INTACT – therefore, the authors may not discuss this anywhere as a comparative result. Therefore the conclusion on line 219 is wrong. I did not understand would anyone want to wait for 24 hrs before processing nuclei, especially if they are not fixed. And this is more confusing as from Table 1 the longest sort was only 5 hours (300 mins).

We have removed the 24 hrs experiments. Initially they were done to observe membrane dynamics. We did not use these nuclei for any other purpose.

  • Section 2.4 – FANS and INTACT are not treatments in the classical sense

up-regulated? down-regulated?  -> better is enriched/depleted or more/less present compared to reference. Aren’t all of the samples from the same biological source? There is no up or down really so the expression should be changed and please make sure the reference sample is clearly indicated. I think sometimes it was, sometimes it wasn’t. If it will be consistent – make a statement first which the reference will be, and then continue.

As suggested by the reviewer we have modified the terms to enriched/depleted.

Reviewer 2 Report

In this manuscript, Chongtham et al. present a side-by-side comparison of different nuclei purification methods for downstream molecular analysis. The authors first provide an overview of the trend of methods used to isolate nuclei in the recent years, and then provide comparative analysis of effects of these different nuclei isolation methods on downstream molecular analysis. The conclusion is that the flow cytometer based isolation (FANS) provide better purity without serious optimization, but has potential artificial effects on chromatin accessibility as well as RNA content due to various stress during the sorting, and also difficulty in scaling up the number of samples. On the other hand, the magnetic beads based enrichment allows easier multiple sample handling with less harm, while requiring optimization for the bead enrichment step for different amounts of input. The difference in the molecular profiles between the two techniques are intriguing, and would like to have more reasonable explanations for this observation.

Overall, the conclusion is reasonable and the manuscript is well written. I would expect that researchers across broad areas will benefit from this paper. Therefore, I recommend the manuscript to be accepted with the addition of a more in-depth description on the potential cause of differential molecular profiles between the two isolation protocols. A careful check on the typos (stylistic) should also be carried out before publication/

Minor point

As mentioned above, the difference in RNA-seq profiles and ATAC-seq profiles are surprising. I would assume that once the nuclei is isolated, new transcription would not occur due to diluted NTPs and other components required. However, the authors refer to changes in chromatin accessibility upon stree could lead to transcriptional activity (line 470-472). If transcription would occur in isolated nuclei (which I feel unlikely), would the addition of a transcriptional inhibitor minimize the difference between INTACT and FANS? Has this type of analysis been done in nuclear-RNA seq methods?

In addition, if the authors perform a motif enrichment analysis of differential accessibility sites, would it be possible to identify factors that respond to stress conditions?

Author Response

Reviewer 2

In this manuscript, Chongtham et al. present a side-by-side comparison of different nuclei purification methods for downstream molecular analysis. The authors first provide an overview of the trend of methods used to isolate nuclei in the recent years, and then provide comparative analysis of effects of these different nuclei isolation methods on downstream molecular analysis. The conclusion is that the flow cytometer based isolation (FANS) provide better purity without serious optimization, but has potential artificial effects on chromatin accessibility as well as RNA content due to various stress during the sorting, and also difficulty in scaling up the number of samples. On the other hand, the magnetic beads based enrichment allows easier multiple sample handling with less harm, while requiring optimization for the bead enrichment step for different amounts of input. The difference in the molecular profiles between the two techniques are intriguing, and would like to have more reasonable explanations for this observation.

Overall, the conclusion is reasonable and the manuscript is well written. I would expect that researchers across broad areas will benefit from this paper. Therefore, I recommend the manuscript to be accepted with the addition of a more in-depth description on the potential cause of differential molecular profiles between the two isolation protocols. A careful check on the typos (stylistic) should also be carried out before publication/

We thank the reviewer for these comments. As suggested by reviewer 1 we have now modified the manuscript to focus on both, similarities and differences in the molecular profiles between the two isolation techniques.

Minor point

As mentioned above, the difference in RNA-seq profiles and ATAC-seq profiles are surprising. I would assume that once the nuclei is isolated, new transcription would not occur due to diluted NTPs and other components required. However, the authors refer to changes in chromatin accessibility upon stree could lead to transcriptional activity (line 470-472). If transcription would occur in isolated nuclei (which I feel unlikely), would the addition of a transcriptional inhibitor minimize the difference between INTACT and FANS? Has this type of analysis been done in nuclear-RNA seq methods?

Although it was shown that both transcription and structural modifications in the chromatin still occur in isolated nuclei, we are not aware of any study that used a transcriptional inhibitor before nucRNA-seq. Our aim here was to report the advantages/disadvantages of each isolation technique.

In addition, if the authors perform a motif enrichment analysis of differential accessibility sites, would it be possible to identify factors that respond to stress conditions?

As suggested we have now performed motif enrichment analysis and found six motifs that are associated at promoters of FANS-gained accessibility regions. We have included the results of this analysis in section 2.5.

Reviewer 3 Report

GENERAL COMMENTS

The article by Chongtham presents a comparison between two pre-analytical methods for single nuclei sorting: i) INTACT and ii) FANS. Authors hypothesized that technical differences in these two approaches would lead to different results when nuclei are using in RNA-Seq and ATAC-Seq. The authors started by reviewing the available data on both technologies, followed up by characterization of technical performance (i.e. sorting speed and efficiency) and comparative RNA-Seq and ATAC-Seq analyses.

In principle, the idea of the manuscript is sound. It is widely accepted that pre-analytic methods in single nuclei analysis may impact the downstream results. However, in order to obtain conclusive results such study will require to follow existing guidelines for method verification, including multiple samples per method, as well as the repetition using different batches, days, operators, etc. As this study includes a limited number of samples, the results are purely descriptive and, in this reviewer's opinion, do not provide enough evidence supporting the characterization of the advantages and pitfalls of each method.

MAJOR POINTS

  • Results: Table 1. Were this observation made only once? Are the reported mean or median values, what is the standard deviation? Also, providing the theoretical values as ">60" is imprecise and cannot be used to support the statement that INTACT is faster than FANS.
  • Results: Table 1: Are the differences in efficiency statistically significant?
  • Results: What is the rationale of selecting 3 and 24 hours post-sorting as time points. To this reviewer's experience is highly recommended that the libraries are prepared right after nuclei separation and this analysis does not seems very relevant.
  • Results: There are only two biological replicate per group. Although it is acknowledge that the library preparation is expensive, the number of biological replicates appears low for drawing any conclusion.
  • Results: The RIN values in groups 2 and 3 partially overlap (see Supp. Fig.4). What is the rationale to consider these samples as two independent groups instead of a single one (i.e. RIN < 7)?
  • Results: The overall PCA analysis discriminating by RIN-group and sorting technique needs to be revised. First, for the sentence "PCA of all the samples discriminated with respect to the batch of isolation (Supplementary Fig. 4A).", Supplementary Fig. 4A is a table containing the RIN values. Apparently, the results of this analysis are in Supp. Fig. 5. Further down the text, the authors claim that "Individual PCA analysis of groups 2 and 3 was able to discriminate the samples based on technique while revealing more distinct transcriptome patterns of FANS- nuclei compared to INTACT-nuclei (Supplementary Fig. 4C-D), similar to our previous observations of the high-quality RIN group (Supplementary Fig. 4B)." This is confusing, the PCA analysis show clear discrimination in all groups (Fig 5B, C and D).
  • Results: How many biological replicates were used in the ATAC-Seq experiment? Are the differences between the groups statistically significant?

MINOR POINTS

  • Abstract: Correct the sentence "These nuclei are then used for molecular and epigenetic characterization of the cellular sub-populations.". Epigenetic characterization is in fact a molecular characterization.
  • Results: The authors refer as "meta analysis" to the review and summarization of publications using FANS and INTACT. This is not truly a meta-analysis, since does not involve the grouping and re-analysis of data from independent studies but rather the counting of observations. The review is very well performed, but I would suggest not referring to it as "meta-analysis" for accuracy of technical terminology.
  • Results: When describing the results of table 1, please add the numbers in the main text as well.

Author Response

Reviewer 3

The article by Chongtham presents a comparison between two pre-analytical methods for single nuclei sorting: i) INTACT and ii) FANS. Authors hypothesized that technical differences in these two approaches would lead to different results when nuclei are using in RNA-Seq and ATAC-Seq. The authors started by reviewing the available data on both technologies, followed up by characterization of technical performance (i.e. sorting speed and efficiency) and comparative RNA-Seq and ATAC-Seq analyses.

In principle, the idea of the manuscript is sound. It is widely accepted that pre-analytic methods in single nuclei analysis may impact the downstream results. However, in order to obtain conclusive results such study will require to follow existing guidelines for method verification, including multiple samples per method, as well as the repetition using different batches, days, operators, etc. As this study includes a limited number of samples, the results are purely descriptive and, in this reviewer's opinion, do not provide enough evidence supporting the characterization of the advantages and pitfalls of each method.

We thank the reviewer for these comments. We feel that the description of the samples used for the analyses might have been a bit misleading. The Supplementary Figure 4A contains additional information on the number of biological replicates, batches and technical replicates per technique. Overall our study contains 32 nucRNA-seq samples in total, derived from 3 different batches with 4-7 replicates per technique.

MAJOR POINTS

  • Results: Table 1. Were this observation made only once? Are the reported mean or median values, what is the standard deviation? Also, providing the theoretical values as ">60" is imprecise and cannot be used to support the statement that INTACT is faster than FANS.

The observations were mean values of duplicate replicates. We have now changed the wording and in the new version of the manuscript do not claim that INTACT is faster than FANS but only suggest that this could be the case.

  • Results: Table 1: Are the differences in efficiency statistically significant?

We have changed efficiency into yield to avoid confusion. As we are reporting yield of sfGFP+ve INTACT nuclei from a mean of duplicates, statistical significance is not considered. We decided to report the yield for only INTACT-nuclei as INTACT-nuclei sorting is highly dependent on the sfGFP expression on the nuclear membrane. This expression can be highly dependent on the tissue regions.

  • Results: What is the rationale of selecting 3 and 24 hours post-sorting as time points. To this reviewer's experience is highly recommended that the libraries are prepared right after nuclei separation and this analysis does not seems very relevant.

We did not use the 24hrs samples for sequencing and have removed them from the manuscript. Although we started imaging the nuclei directly after sorting, capturing all images necessary took up to three hours. 

  • Results: There are only two biological replicate per group. Although it is acknowledge that the library preparation is expensive, the number of biological replicates appears low for drawing any conclusion.

In this analysis we wanted to test and compare the library preparations. This comparison has been done previously on total and nuclear RNA (Ref 47-49), however we were interested if there were some differences between the library prep (Nugen/Clontech) in combination with specific technique (INTACT/FANS). Overall, we see that the samples cluster according to the library prep (PC1) and technique (PC2). This is a type of control experiment to track the differences between these conditions and which library preparation is more appropriate for both of the techniques and type of material (nuclei). Therefore, we think that two technical replicates (from the same biological source) are sufficient for the comparison.

  • Results: The RIN values in groups 2 and 3 partially overlap (see Supp. Fig.4). What is the rationale to consider these samples as two independent groups instead of a single one (i.e. RIN < 7)?

We are sorry for the confusion. We now mention in the text that “…we compared the different batches and analyzed the samples according to the isolation technique. PCA of all the samples discriminated with respect to the batch of isolation (Supplementary Fig. 4A).” Additionally, we modified section 2.4 to be more clear as suggested by reviewer 1. To avoid confusion, we changed the term “groups” to “batches” (1,2,3) to be clear about the source of the material, the processing, and the RNA quality.

  • Results: The overall PCA analysis discriminating by RIN-group and sorting technique needs to be revised. First, for the sentence "PCA of all the samples discriminated with respect to the batch of isolation (Supplementary Fig. 4A).", Supplementary Fig. 4A is a table containing the RIN values. Apparently, the results of this analysis are in Supp. Fig. 5. Further down the text, the authors claim that "Individual PCA analysis of groups 2 and 3 was able to discriminate the samples based on technique while revealing more distinct transcriptome patterns of FANS- nuclei compared to INTACT-nuclei (Supplementary Fig. 4C-D), similar to our previous observations of the high-quality RIN group (Supplementary Fig. 4B)." This is confusing, the PCA analysis show clear discrimination in all groups (Fig 5B, C and D).

We are sorry for this mistake. In the new version of the manuscript we have corrected this. In addition, as suggested by reviewer 1 we have modified section 2.4 and have linked the text and supplementary information correctly.

  • Results: How many biological replicates were used in the ATAC-Seq experiment? Are the differences between the groups statistically significant?

We included the number of replicates used for comparison (3 technical replicates) in the text. Using DESeq, the significant differentially accessible regions (DARs) are calculated (p-value cutoff of 0.05). As a result, also the identified GO terms and motif enrichment analysis are associated with the significant DARs.

MINOR POINTS

  • Abstract: Correct the sentence "These nuclei are then used for molecular and epigenetic characterization of the cellular sub-populations.". Epigenetic characterization is in fact a molecular characterization.

Thank you for this correction. We have changed the text accordingly.

  • Results: The authors refer as "meta analysis" to the review and summarization of publications using FANS and INTACT. This is not truly a meta-analysis, since does not involve the grouping and re-analysis of data from independent studies but rather the counting of observations. The review is very well performed, but I would suggest not referring to it as "meta-analysis" for accuracy of technical terminology.

We have now changed the description from “meta analysis” to “literature review”.

  • Results: When describing the results of table 1, please add the numbers in the main text as well.

We have now added the numbers in the main text.

Round 2

Reviewer 1 Report

Personally, I still think the authors are likely just looking at noise. But that is what science is for, to put your ideas out there, some will hold water, some will sink. But, as far as the publication goes, I think we have improved it enough so that the science and observational bias has been much reduced from its initial form.

The main text is fine, but very difficult to read with all the markings all over the place. I recommend that the authors read it carefully without them, and try to catch any other small errors that might be hiding. For example, it is impossible to actually judge Table 1. I suppose this is more a comment at MPDI than the authors.

Supp. fig legends 2 and 3 are swapped. See other comments in supp file.

Author Response

Reviewer 1

Personally, I still think the authors are likely just looking at noise. But that is what science is for, to put your ideas out there, some will hold water, some will sink. But, as far as the publication goes, I think we have improved it enough so that the science and observational bias has been much reduced from its initial form.

The main text is fine, but very difficult to read with all the markings all over the place. I recommend that the authors read it carefully without them, and try to catch any other small errors that might be hiding. For example, it is impossible to actually judge Table 1. I suppose this is more a comment at MPDI than the authors.

We went through the manuscript carefully and corrected the errors.

Supp. fig legends 2 and 3 are swapped. See other comments in supp file.

We are sorry for this mistake. The new version of the manuscript contains the correct figure legends.

Reviewer 3 Report

Authors are commended for the efforts. The manuscript is somehow improved now, however, the main issue remains. The sample number per group is still very low to be able to draw any conclusions. Especially, if two technologies are compared, experimental design must adjust to best practices and guidelines to ensure accuracy and robustness of the results.

Unless the sample number per group is considerable increased or a power analysis is provided in which the current number of biological and technical replicates is supported, I don't think the manuscript is not deemed for publication.

Author Response

Reviewer 3

Comments and Suggestions for Authors

Authors are commended for the efforts. The manuscript is somehow improved now, however, the main issue remains. The sample number per group is still very low to be able to draw any conclusions. Especially, if two technologies are compared, experimental design must adjust to best practices and guidelines to ensure accuracy and robustness of the results.

Unless the sample number per group is considerable increased or a power analysis is provided in which the current number of biological and technical replicates is supported, I don't think the manuscript is not deemed for publication.

We thank the reviewer for this comment. In our opinion, however, a proper comparison between INTACT and FANS required similar input material which in our case were nuclei originating from same biological source (neuronal population). We believe that technical replicates provided a more informative result for molecular comparison of the two techniques than biological replicates.